# Entero-Cutaneous and Entero-Atmospheric Fistulas: Insights into Management Using Negative Pressure Wound Therapy

**DOI:** 10.3390/jcm13051279

**Published:** 2024-02-23

**Authors:** Gilda Pepe, Maria Michela Chiarello, Valentina Bianchi, Valeria Fico, Gaia Altieri, Silvia Tedesco, Giuseppe Tropeano, Perla Molica, Marta Di Grezia, Giuseppe Brisinda

**Affiliations:** 1Emergency Surgery and Trauma Center, Department of Abdominal and Endocrine Metabolic Medical and Surgical Sciences, Fondazione Policlinico Universitario A. Gemelli, IRCCS, Largo Agostino Gemelli 8, 00168 Rome, Italy; gilda.pepe@policlinicogemelli.it (G.P.); valentina.bianchi@guest.policlinicogemelli.it (V.B.); valeria.fico@policlinicogemelli.it (V.F.); gaia.altieri@guest.policlinicogemelli.it (G.A.); silvia.tedesco55@gmail.com (S.T.); giuseppe.tropeano@guest.policlinicogemelli.it (G.T.); perla_molica@hotmail.it (P.M.); marta.digrezia@policlinicogemelli.it (M.D.G.); 2General Surgery Operative Unit, Department of Surgery, Provincial Health Authority, 87100 Cosenza, Italy; michela.chiarello@aspcs.it; 3Department of Medicine and Surgery, Catholic School of Medicine “Agostino Gemelli”, Largo Francesco Vito 1, 00168 Rome, Italy

**Keywords:** enteric fistula, enterocutaneous fistula, entero-atmospheric fistula, negative pressure wound therapy, open abdomen

## Abstract

Enteric fistulas are a common problem in gastrointestinal tract surgery and remain associated with significant mortality rates, due to complications such as sepsis, malnutrition, and electrolyte imbalance. The increasingly widespread use of open abdomen techniques for the initial treatment of abdominal sepsis and trauma has led to the observation of so-called entero-atmospheric fistulas. Because of their clinical complexity, the proper management of enteric fistula requires a multidisciplinary team. The main goal of the treatment is the closure of enteric fistula, but also mortality reduction and improvement of patients’ quality of life are fundamental. Successful management of patients with enteric fistula requires the establishment of controlled drainage, management of sepsis, prevention of fluid and electrolyte depletion, protection of the skin, and provision of adequate nutrition. Many of these fistulas will heal spontaneously within 4 to 6 weeks of conservative management. If closure is not accomplished after this time point, surgery is indicated. Despite advances in perioperative care and nutritional support, the mortality remains in the range of 15 to 30%. In more recent years, the use of negative pressure wound therapy for the resolution of enteric fistulas improved the outcomes, so patients can be successfully treated with a non-operative approach. In this review, our intent is to highlight the most important aspects of negative pressure wound therapy in the treatment of patients with enterocutaneous or entero-atmospheric fistulas.

## 1. Introduction

Enteric fistulas (EF) are challenging conditions in abdominal and gastrointestinal surgery and remain associated with significant morbidity and mortality rates, due to complications such as sepsis, malnutrition, and electrolyte imbalance [1,2,3,4,5,6].

EFs are most commonly iatrogenic and usually the result of a surgical misadventure (e.g., anastomotic leakage, injury of the bowel or blood supply) [7]. In addition, fistulas may result from erosion by suction catheters, adjacent abscesses or trauma. Contributing factors may include previous chemo-radiation therapy, intestinal obstruction, inflammatory bowel disease, mesenteric vascular disease, or intra-abdominal sepsis [8,9,10,11].

The abnormal fistulous connection can occur between the gastrointestinal tract and the skin (enterocutaneous fistula, ECF), another portion of the gastrointestinal tract (entero-enteric fistula) or the urogenital tract. The increasingly widespread use of open abdomen (OA) techniques for the initial treatment of abdominal sepsis and trauma has led to the observation of the so-called entero-atmospheric fistulas (EAF) [12,13,14,15].

Because of their clinical complexity, proper management of EF requires a multidisciplinary team. The main goal of the treatment is the closure of the fistula. Mortality reduction and improvement of a patient’s quality of life are also fundamental. The treatment must include correction of malnutrition, which can be achieved both through parenteral and enteral nutrition even for a long time, and surgical strategies to obtain abdominal closure [16,17,18].

ECF and EAF show different characteristics and require different approaches. The resolution rate of ECF with conservative management is between 19.9 and 81.4% [19,20]. With the wound care measures used, 90% of spontaneous closures occurred in the first month after the resolution of sepsis. An additional 10% of closures occurred in the second month. No spontaneous closure occurred after 2 months [21]. With vacuum-assisted closure (VAC) and other negative pressure wound therapies (NPWT), there are case reports of fistulae closure during the second and third months [22,23]. Surgery, instead, is needed for 13–80% of the cases and it is influenced by anatomical and physiological features. This heterogeneity determines different treatments and a lack of a standardized protocol [24]. In EAF, the first-choice treatment is conservative, and surgical therapy should be considered only in case of its failure because of the high risk of bowel injury due to the presence of a “frozen abdomen” [15,25,26,27,28,29,30].

Despite the advances in perioperative care and nutritional support, the mortality risk of fistulas remains between 15 and 30% [31,32,33]. In more recent years, the use of negative pressure wound therapy (NPWT) for the spontaneous resolution of EF improved the outcomes, so patients can be successfully treated with a non-invasive approach [34,35]. In this review, our intent is to highlight the most important aspects of NPWT in the treatment of patients with EFs.

## 2. Definition of Fistula

A fistula is an abnormal communication between two epithelial surfaces. Fistulas are defined as ECF when there is communication between a bowel segment and the skin surface (Figure 1).

EAF is a peculiar type of ECF. Because of the lack of fascia, soft tissues, and/or skin surrounding them, they are characterized by a hole in the bowel, which is in contact with the atmosphere, directly (Figure 2).

The internal fistulas, instead, are characterized by a gut-to-gut connection.

## 3. Etiology of ECF

Surgery remains the most common cause of ECF due to anastomotic leakage or an unrecognized injury of the bowel [36]. About 50% of fistulas are due to non-anastomotic iatrogenic perforations. Another 50% is due to a partial or total anastomotic dehiscence [36,37]. Patients who undergo a challenging adhesion dissection have a greater risk of iatrogenic perforation.

An intestinal infection can determine an abscess formation and create an internal and/or external fistula. These fistulas are quite common in the developing countries. Typhus is the most common cause of post-infective fistulas, followed by salmonellosis, amebiasis, actinomycosis, coccidiomycosis, and HIV. Also, abscesses due to complicated appendicitis or diverticulitis can cause entero-enteric, entero-vesical, entero-vaginal or ECFs.

Fistulas are common complications in Crohn’s disease (20–40%) [38]. The terminal ileum is the tract more frequently involved. Chronic inflammations determine adhesions among the structures, and this can cause micro-abscess, ulcer formation or even perforation.

The post-radiotherapy fistula usually occurs in the first 5 years after radiant therapy, but can also happen 30 years later, more rarely. The physio-pathological mechanism consists of persistent damage caused to the gut’s vessel and connective tissue that can determine ischemia, erosion, and a fistula because of the tenacious adhesions. Ablation with radiofrequency or other methods (e.g., microwave) can directly cause damage to nearby organs. Ablative therapies used in liver or kidney tumors can cause the appearance of intestinal perforation due to direct, thermal or chemical injury, with subsequent formation of abscesses and fistulas or, in severe cases, peritonitis [39,40]. In the literature, 4–8% of patients treated with abdominopelvic radiation therapy can develop serious complications such as fistulas, perforation, or abscesses [41,42]. Radiation dose, radiation field size, and combination with other anticancer treatments are the main risk factors for intestinal perforation during radiotherapy [43]. Several studies describe cases of intestinal perforation following the use of radiotherapy together with antiangiogenetic agents, like dabrafenib and trametinib for pelvic bone melanoma metastases [44], sorafenib in renal cancer patients [45], and gefitinib in a patient with lung cancer receiving lumbar irradiation [46].

Bowel perforation is a rare but serious complication of chemotherapy [11]. Several mechanisms may be responsible for gastrointestinal perforation from oncologic treatments. Anticancer drugs induce vascular damage by thrombosis and thromboembolism. In these cases, when intestinal vessels are involved, bowel ischemia with perforation may occur [47]. Perforation of the gastrointestinal tract can also occur after prolonged obstruction [48] or due to treatment responses with tumor lysis, as in cases of lymphomas or gastrointestinal stromal tumors [49]. Finally, bowel perforation can be a result of other complications of oncologic therapies, like pneumatosis or enterocolitis. Various chemotherapeutic agents can cause perforation. In the literature, the drugs with which this complication most commonly occurs are fluorouracil, taxols, cisplatin, interleukin-2, and mitomycin [50,51,52,53]. Among the molecular targeted therapies, bevacizumab is most commonly associated with gastrointestinal perforation, with an incidence of 0.9% [54] and a correlation with late anastomotic leakage [55]. Risk factors for bevacizumab-related perforation are specific tumors (colorectal, prostate, and gynecological cancers), combination with other treatments, such as oxaliplatin and taxanes, the presence of a primary tumor in situ, and a recent history of endoscopy or abdominal radiotherapy [54,56,57,58]. Bowel perforation occurs in 80% of patients during the first 6 months after bevacizumab administration [59] and the most common sites of perforation are the colon, small intestine, and stomach, respectively [60]. The pathophysiological mechanisms underlying intestinal perforation from molecular target therapy are different. With the use of these drugs, an antiangiogenic action occurs. This condition reduces capillary density at the mucosal level, which can compromise the integrity of the intestinal wall. Furthermore, tumor lysis, in response to treatment, the increased risk of thromboembolic events in the mesenteric vessels and the regression of normal blood vessels contribute to the appearance of perforation [61,62]. Several studies show an association between gastrointestinal perforation and antiangiogenic tyrosine kinase inhibitors, like erlotinib, regorafenib, sunitinib, and sorafenib [63,64,65,66,67,68].

## 4. EAF and OA

EAFs are a devastating complication of OA treatments and originate when there is a large dehiscence of the abdominal wall or in the case of a “frozen abdomen” [15,30,69,70]. EAFs are associated with severe morbidity rates, reduced patient health-related quality of life, and increased rates of mortality, duration of hospital stay, and hospital costs [71]. The cost of fistula care is significant and typically more than $500,000 [72].

Damage control surgery for abdominal trauma and/or for source control in severe abdominal sepsis and the management of abdominal compartment syndrome are the main clinical conditions in which OA (Figure 3) is currently applied [12,17,27,73,74,75,76,77,78,79,80].

The etiology of an EAF may often be multifactorial and represent a combination of several independent factors including the primary diagnosis and cause for OA treatment, iatrogenic lesions of the intestinal tract during laparostomy/relaparotomy, postoperative anastomotic rupture, dehydration, swelling and ischemia of the intestine, exposure of the bowel to materials used for temporary abdominal closure, adhesions between the bowel and the abdominal wall, wound infections. Delayed abdominal closure, large fluid resuscitation volume (>5 L/day) and the application of polypropylene mesh directly to the viscera increase the risk of this complication [17]. Although the use of NPWT, presence of bowel injury, bowel repairs or anastomoses, and intra-abdominal infection/sepsis were previously considered risk factors for EAF, subsequent studies did not support these associations [77,81]. The duration of OA management and poor nutritional status seem to be the most important predictors of EAF formation in patients with OA [81].

In trauma patients treated with OA, the incidence of EAFs has been associated with large-volume resuscitation and an increasing number of re-explorations [82]. NPWT, also used to treat an EAF, is linked to their development in 5% of patients. Earlier reports have revealed the incidence of EAFs to approximate 20% during NPWT.

The incidence of EAF varies between 2% and 25% for trauma patients, 20% and 25% for abdominal sepsis, and up to 50% for infected pancreatic necrosis, with a mortality rate of 20% to 44% [83,84].

Clinically, EAFs are usually seen after the first week of OA, but they can occur at any time during the hospitalization, especially in patients requiring many months of OA [85,86].

## 5. Classification and Evolution

A fistula is often classified according to output, etiology, and source. These factors dictate both treatment and morbidity/mortality rates. ECF can be internal and external fistulas. This helps in identifying the organs involved and providing characteristics of the fistula tract. Internal fistulas have communications between two hollow viscera and if symptomatic should be treated by resection and re-anastomosis. The external fistulas connect hollow viscera to the skin. External fistulas that have favorable outcomes include esophageal, duodenal stump, pancreaticobiliary, and jejunal fistulas with small enteric defects (<1 cm) and long tracts (>2 cm). Gastric, lateral duodenal, ligament of Treitz, and ileal fistulas are less favorable to close spontaneously (Figure 4).

According to the organ of origin, ECF can be classified into type I (abdominal, esophageal, gastroduodenal), type II (small bowel), type III (large bowel), and type IV (EAF, regardless of origin). Small bowel fistulas (type II) are more common (60 to 70%) than fistulas from other parts of the gastrointestinal tract. EAFs can also be classified as proximal or distal, or, in OA, as superficial or deep.

The classification according to the output is very useful when we opt for a conservative treatment. EFs may be classified as low (<200 mL/day), moderate (200–500 mL/day), and high (>500 mL/day)-output fistulas [87]. Generally, the more proximal the fistula is located within the gastrointestinal tract, the higher the output.

Fistulas can be classified as either simple or complex. The simple fistulas have a small track and are not associated with abscesses. Moreover, they do not involve other organs. On the other hand, the complex fistula can be grouped into type I, if a fistula determines an abscess and involves different organs, or type II if the bowel hole opens in a disrupted wound and there is neither soft tissue nor skin. Complex fistulas have high mortality and morbidity rates, and they are unlikely to reach a spontaneous resolution.

Location and fistula output affect the prognosis and spontaneous closure of the ECF/EAF crucially (Table 1).

Generally, distal and low-output EFs have a high spontaneous closure rate in contrast to more proximal and high-output fistulas. Low-output fistulas have a higher possibility of spontaneous resolution than high-output fistulas [20,88]. Furthermore, the location of the EAF deep in the peritoneal cavity is considered a surgical emergency due to ongoing peritonitis. The deep EAF should therefore be managed immediately. On the contrary, superficial EAFs occurring on the granulated surface of the OA are more frequently encountered in clinical practice. To be sure, they are more difficult to manipulate due to the inability to control the enteric spillage on the OA surface, effectively. This also triggers ongoing sepsis.

The persistence of ECF may be secondary to additional factors [89]. The spontaneous closure is unlikely to happen in case of abscesses, circular involvement of the bowel segment, hypoperfusion of the intestinal wall, short fistula tract (<2 cm), bowel wall injury >1 cm, diseased bowel, foreign bodies or distal obstruction.

## 6. Physiopathology

ECF causes a loss of fluids, electrolytes, and nutrients that can determine or worsen dehydration, electrolyte imbalance, malnutrition, skin, and wound lesions (e.g., abrasions, mycosis, bacterial infections). All these conditions can enhance and make patients more vulnerable to hypovolemia, arrhythmias, immunodeficiency, asthenia, pneumonia, sepsis, liver and renal failure, shock, and death in the most severe cases [90,91].

High-output fistulas are a risk factor for intestinal failure because of a massive reduction of absorption [92]. The increased intestinal losses of fluids and electrolytes, the disruption of the enterohepatic cycle, the restricted oral/enteral nutrition or total fasting (bowel rest) to decrease fistula output, the impaired intestinal peristalsis, and increased metabolic demand related to concomitant sepsis and inflammation are further mechanisms of intestinal failure [93]. So, a proper supplement of water, electrolytes, macro-nutrients, and total parenteral nutrition is necessary for this purpose [16]. In the case of low and moderate output, instead, an oral supplement of nutrients is usually enough.

## 7. Outcomes

The mortality rate is currently about 5.3–21.3%, despite a previous much higher rate of 40–65% [20,94,95]. This reduction is probably due to improvements in nutritional support, intensive care, antimicrobial strategies, and surgical invasive and non-invasive techniques (i.e., NPWT). The main causes of death in these patients continue to be electrolyte imbalance, severe malnutrition, and sepsis. This is particularly true for patients affected by high-output fistulas, such as duodenal and jejunal fistulas. The mortality rate still reaches 35% in these cases [96]. Hypoalbuminemia (albumin < 3.5 g per 100 mL) is most commonly associated with fistula formation [32,97]. Patients with hypoalbuminemia have increased morbidity and mortality rates associated with fistula formation [98].

Over the past few decades, mortality rates for EAF have generally decreased from more than 70% to 40%, as a result of more appropriate advanced intensive care and improved surgical techniques [1,15,17,99,100]. The morbidity rates of EAFs are also high. These are related to the pathophysiological consequences. Changes that occur in EAF include severe fluid and electrolyte losses, acid–base homeostasis imbalance, hypercatabolism (hypoalbuminemia and hypoproteinemia), vitamin and trace element deficiencies, and septic wound complications due to leakage of enteric effluents onto the open abdominal surface.

However, after the closure of a fistula, recurrence may occur. This event has an incidence of 9−33% in patients with spontaneously sealed EAF. The EAF severity, in terms of morbidity and mortality, is reflected in Bjorck’s classification [101] of the OA, where an OA with an EAF is considered to be grade 3, a step just before the frozen and inoperable OA (grade 4).

## 8. Treatment

There are some defined principles that must be followed for the management of ECF and EAF. A common acronym used to describe ECF care protocol is “SNAP”, which stands for the management of skin and sepsis, nutrition, the definition of fistula anatomy, and proposing a procedure to address the fistula [102,103].

Sepsis control and an adequate nutritional status have to be reached in order to obtain a fistula closure [89,104,105]. Sepsis is responsible for 77% of mortality associated with ECF [3]. A bowel obstruction in a distal intestinal segment does not allow the fistula to heal. Spontaneous closure in patients affected by malignancies or Crohn’s disease is infrequent.

The management of patients with an OA and an EAF is very challenging [18,97]. These patients are usually critically ill and hypercatabolic and deteriorate rapidly if complications occur during their hospitalization in the intensive care unit. Unfortunately, proximal diversion (Figure 5) of enteric contents is almost technically impossible to achieve, due to the thick and shortened mesentery, the edematous bowel, the noncompliant abdominal wall, and the rather hostile environment of the OA.

The principles of the non-operative support of the critically ill patient with an EAF are the following:(a)Recognition and management of sepsis: organ dysfunction or progressive organ failure should be promptly managed;(b)Source control is crucial for the resolution of sepsis;(c)Antibiotic regimen based on culture results;(d)Reducing fistula output: nil per os, a nasogastric tube as well as an attempt to reduce secretions of the gastrointestinal tract by administering proton pump inhibitors and to reduce enteric and pancreatic secretions using somatostatin or octreotide [106];(e)Nutritional support: In the presence of OA, the patient is in a hypercatabolic state. Hypercatabolism is further worsened by the presence of an EAF. The main parameters on which nutritional support should be based are the following: (a) increased caloric requirements, usually calculated by 30–35 kcal/kg/day; (b) increased protein depriving, calculated by adding 1.5 g protein/kg/day and 2 g protein losses for each liter of fluid collected from the raw surface of the OA; and (c) deficiencies of vitamins and trace elements. Adequate nutritional support based on the patient’s nutritional status, a positive nitrogen balance, adequate trace minerals, and vitamin replacement, along with glycemic control, may allow the surgeon to proceed to surgical treatment of the fistula. Additionally, several known parameters, such as weight, prealbumin, albumin, and transferrin, are correlated with postoperative mortality and morbidity and spontaneous fistula closure rates [92,107,108].

Once sepsis and electrolyte imbalance have been treated, defining the fistula anatomy is mandatory. So, a CT scan, a fistulography or a bowel/colonic transit have to be performed. Studying the rectum or colon patency is also useful, especially in those patients who had a colostomy or diverticulitis.

The operating surgeon must define the best surgical approach and technique for closing the fistula. Moreover, it is also important to select the right mesh to use in case of abdominal hernias or if the fascial gap is too wide. The main goal of the surgical approach is to improve the patient’s quality of life. Other targets are recreating bowel continuity, minimizing ECF recurrence risk, and avoiding hernias and surgical site infection. Sometimes, different techniques have to be combined to reach all the above-mentioned outcomes. Patients with ECF or EAF have an abdomen that can be defined as “hostile”, and access can be challenging for the surgeon (Figure 6).

Reconstructive surgery should be postponed for 3–6 months [109]. Spontaneous healing is possible for ECF, but most of them, especially in cases of high output, require surgical treatment. The time interval in which operative intervention is associated with significantly higher mortality has been outlined in the literature. Return to the operating room within 10 days resulted in 13% mortality, operating between 11 and 42 days was associated with 21% mortality, and, after 42 days, mortality returned to 11% [110]. The absolute minimal waiting interval after the original surgery is 6 weeks.

The new laparotomy should be as far as possible from the previous incisional site, but this is usually not possible (especially for patients who had an urgent laparotomy for trauma). Also, a transverse incision can be an option. It is necessary to excise the involved bowel loop for making a fistula excision. If there are more fistulas, the ideal scenario is when they are all in continuous and closed segments. So, a single en-bloc resection and anastomosis can be performed. All of the bowel and colon must be explored, from the Treitz ligament to the rectum. If there are doubts about the anastomosis, it needs to be revised or a loop ileostomy can be performed.

The surgical treatment of ECF has a successful rate of about 80–90% [89,95,111,112]. It is crucial not to cause other perforations during the dissection. Surgery principles of avoiding a too extensive resection of the mesentery and tension on the anastomosis or performing it on a hypoperfused tract are still valid. The failure rate increases with infectious and noninfectious complications. Additionally, following surgery, ECF recurrence is 14 to 34% [113,114]. Recurrence rates are minimized (18%) when the involved bowel is fully mobilized and resected. Oversewing or wedge resection/bowel repair results in higher rates of recurrence at 33%. Similarly, Runström et al. reported that the ECF failure rate is lower when no anastomosis is constructed and, instead, a stoma is chosen (recurrence rate 14 vs. 34% with anastomosis) [114]. More broadly, one has to balance the risk of ECF recurrence with the morbidity of another operation if the anastomosis is avoided.

The resection of the involved enteric loop is actually the most definite way of treating an EAF [97,115,116,117]. However, this therapeutic option is only feasible in clinically stable patients, in good nutritional conditions, and, above all, in conditions that are free of infections. These conditions occur in some patients already after 1–2 months, while in others after 6–12 months or even after a period of 1 year. Regarding the operative strategy, it appears more appropriate to access the peritoneal cavity not directly through the OA but rather through lateral incisions, away from the granulating surface of the OA. Several reports have been published. One of the first reports was by Demetriades, approaching the abdominal cavity via long vertical incisions approximately 8–10 cm lateral to the open abdominal wound, mobilizing the bowel under direct vision toward the midline, and, finally, resecting the involved loops en-block and reestablishing continuity of the gastrointestinal tract with an entero-enteric anastomosis [118].

## 9. NPWT Assisted Closure

As spontaneous EAF closure is rare and depends on fistula localization and output, new treatment options are needed. For the management of OA, NPWT can be considered the standard of care, but the presence of an EAF complicates management remarkably [119,120,121,122,123,124,125,126]. Therefore, the goal is to keep the wound clean and avoid fecal contamination by isolating the EAF.

The first studies on NPWT were focused neither on physio-pathological effects nor on the determination of ideal pressure levels. Morykwas et al. dealt with that with their research on animal models in 1997 [127]. They produced circular wounds of 2.5 cm diameter on the back of a certain number of pigs. Then, they covered them with a polyurethane sponge (sponge pores 400–600 μm). The duplex was used for measuring blood flow in the subcutaneous tissue and the muscles surrounding the lesion. In the meanwhile, growing levels of depression were applied both continuous and intermittent. The granulation process was evaluated considering the dimensional reduction. They compared the results to a control group whose wounds were treated in a conventional way, with swabs and saline solution. The authors observed a better rate of granulation compared to the control group both with continuous and intermittent NPWT (63.3 ± 26.1% and 103 ± 35.3%, respectively). Moreover, these studies demonstrated that NPWT was associated with a significant reduction of bacterial growth compared to the control group after only 4 days of treatment.

NPWT was shown to limit tissue damage in burns on porcine models, measured as an extension of the lesion itself during the first 12 h. The hypothesis was that NPWT could avoid blood flow reduction removing the secretions and the biochemical mediators.

Damage control surgery has spread a lot in recent years and the OA techniques with it [27,77]. This has determined a growing incidence of EAF. This happens because of the trauma that the bowel loops are subjected to during the OA treatment. The introduction of NPWT in the management of EAFs in OA raised many controversies [128]. It was stated that the use of NPWT directly over the intestinal loops might cause the development of new fistulas [129,130]. Higher depression levels were associated with a consistent bowel blood flow reduction with a consequent increase in bowel ischemia and necrosis. This condition can determine a fistula development.

The application of a specially designed non-adherent layer of NPWT system was a breakthrough element that improved the long-term results in the field of OA treatment significantly [6,13,118,131]. It was proven that NPWT facilitates spontaneous EAF closure, especially those characterized by distal location and low-output fistula [2]. The aim of this strategy, which can also involve NPWT, is to transform the fistula into a controlled stoma (Figure 7).

The first experience of treatment with NPWT for open wounds was described by Fleischmann et al. in 1993 when they treated 15 patients with exposed fractures [132]. The authors, who adopted a self-made system, observed excellent wound cleansing and healing, with a great proliferation of granulation tissue. The same authors described a similar experience in 25 patients with lower limb compartment syndrome and in 313 patients with acute and chronic wound infection [133,134]. The average NPWT duration for compartment syndrome was 12.7 days with 2.1 dressings per patient. Other initial experiences with NPWT were recorded by Muller [135] who treated 300 cases of infected wounds. NPWT reduced the wound area in 84% of cases allowing for faster healing and infection control. Smith et al. [136] described the use of NPWT in 93 patients who were treated with OA for various causes in a retrospective review. They reported encouraging results. The devices, duration, and depression for NPWT were very heterogeneous in these first studies, so it was not possible to define this system technical basis. Other studies that followed described NPWT effects on different types of wounds including flaying injuries, infective sternotomies, and soft tissue lesions, which were then treated with direct suture or skin graft.

The use of NPWT has been described also for skin areas subjected to flap removal. The interposition of a low-adherence membrane such as a paraffin gauze dressing is recommended. NPWT use has been also proposed for burning injuries in association with a skin graft, especially for those parts of the body that present an irregular edge or a deep cavity, such as the perineum, the upper limb, and the axilla. The vacuum maintains the graft on the lesion, and it avoids tissue instability due to exudate filtration. Plikaitis and Molnar [137] described the application of a negative pressure system on a skin graft for the treatment of four patients with a deep scalp wound due to a carcinoma excision or burn. Other studies were published on this system and its effect on wounds considered refractory to the healing process. Achieving a hermetic adhesion of the drape to the skin is crucial for the proper functioning of the device. Obtaining this can be challenging for irregular areas surrounded by wet areas, such as the anus or the vagina, unfortunately. A hydrocolloidal membrane between the drape and the skin can be helpful for this. The device introduced in the US in 1995 can guarantee controlled continuous or intermittent pressure levels between −25 and −200 mmHg. A portable device for outpatients is also available.

ECFs were a contraindication to negative pressure therapy until a few years ago because it was considered to promote enteral secretions keeping the fistula open. In general, the crunch question of NPWT in the treatment of the open abdomen is as follows: Does NPWT help to treat fistulas, or does it cause new fistulas? NPWT is associated with higher EAF rates in some studies [138], but a large observational report by Carlson on 578 patients with open abdomen and NPWT showed no elevated EAF rate [139]. Smaller observational studies report EAF rates in open abdomen treatment using NPWT of 5% to 19% [140,141]. Nevertheless, NPWT has been widely used for the treatment of complicated wounds (Figure 8).

In fact, this is a valid option both for low and high-output fistulas. It allows a rapid output reduction (<500 mL/24 h) and early enteral or even oral nutrition. This reduces morbidity and mortality rates and guarantees a faster fistula resolution. Moreover, NPWT reduces the risk of infection by increasing tissue oxygenation and limiting bacterial growth.

Bobkiewicz et al. [144] demonstrated how the use of NPWT has implemented the fistulas’ healing and spontaneous closure. Sixteen patients with a total of 31 EAF were analyzed. The closure rate was higher for those patients treated with intermitted pressure therapy compared to continuous pressure therapy (70% vs. 57%, respectively).

Willms et al. [145] also suggested that covering the bowel with a visceral protective layer is important because those with an OA and secondary peritonitis had a 2.9% risk of EAF formation when a protective visceral layer was used compared with 26.5% when it was not. Some results of the use of NPWT are reported in Table 2.

Spillage of enteric contents of an EAF on the adjacent OA surface serves as a factor of continuous impairment of the healing process. The spillage aggravates local wound sepsis. It is considered a source of major morbidity, too.

Thus, methods of isolation of bowel secretion have been reported and are under study. These are fundamental in order to protect the OA, prevent ongoing sepsis, estimate fluid loss volume and consistency, and facilitate the nursing of the patient. With the constant advances in the field of NPWT, many modifications of NPWT have been reported, especially designed for OA management complicated with EAF. Different techniques are described for fistula isolation, for example, fistula diversion to a floating stoma, Fistula Vac, Tube Vac, Nipple Vac, and Silo Vac [124,129,146,147]. All these techniques are based on individual combinations of different tools used in stoma care and require creativity as well as patience. At the end, a stoma bag is placed on the NPWT dressing to collect the fistula effluent.

**Table 2 jcm-13-01279-t002:** Data on EAF treatment and patients’ outcomes.

Author	Number of Patients/	EAFs	Fistula Closure	Mortality
Woodfield et al. [148]	3	0	100%	0
Goverman et al. [124]	10	5	60%	40%
Wainstein et al. [88]	91	179	80.2%	16.5%
Verhaalen et al. [149]	8	16	90%	10%
D’Hondt et al. [150]	9	17	100%	11.1%
Wang et al. [125]	11	11	100%	0
Ozer et al. [151]	1	1	100%	0
Yetisir et al. [152]	1	1	0	0
Tavusbay et al. [84]	18	Not reported	55.6%	44.4%
Pepe et al. [1]	8	4	62.5%	0
Heineman et al. [153]	4	4	0	0
Jaguscik et al. [154]	1	1	0	0
Bobkiewicz et al. [144]	16	31	61.3%	9.7%
Miranda et al. [126]	2	2	Not reported	0
Ortiz et al. [155]	31	5	74%	6%
Wirth et al. [156]	3	3	100%	0
Sun et al. [123]	83	Not reported	71.1%	0
Wainstein et al. [30]	77	77	81.8%	9%

Subramaniam et al. [146] reported the first effective method of diverting enteric contents from an EAF to a collection bag (floating stoma). This method was applied in three patients successfully and helped the OA to granulate and the EAF to close in a period of approximately 7–10 months.

The introduction of NPWT and the vacuum pack dramatically changed the therapeutic strategy of patients with OA [136,157]. Goverman et al. [124] reported the use of fistula VAC, a method to divert and isolate enteric effluents from EAFs. This method included the protection of the wound bed of the OA with Xeroform dressings, leaving a hole for the associated enteric fistula opening, and the precise fitting of a polyurethane sponge on top of the dressings with a similar hole matching the enteric opening. After covering the sponge with a polyethylene drape, negative pressure was applied at 75 mmHg, while an ostomy bag was fitted in the opening of the EAF draining and diverting enteric fluids away from the surface of the OA. This method was successful in five patients with EAFs.

A similar “fistula VAC” method was reported in 2010 by Byrnes et al. [158], with the difference that a split-thickness skin graft and a vaseline-impregnated dressing on top were applied on the surface of the OA instead of the Xeroform dressings. Then, a similarly applied VAC and ostomy bag were used to divert enteric contents from the hostile OA. Al-Khoury et al. [159] reported an alternative technique for diverting bowel effluent, the so-called Tube VAC. The EAFs were intubated using Malecot catheters of appropriate sizes, while the surface of the OA was covered by petroleum-impregnated gauzes with a polyurethane sponge on top, through which the Malecot catheters were removed. Application of 100 mmHg negative pressure was used for five EAFs in three patients, effectively.

A new method was described for the management of EAFs, consisting of applying nipples over the enteric fistula openings, which were surrounded by VAC WhiteFoam and GranuFoam dressings, effectively diverting enteric effluent into collection bags (nipple VAC) [160]. In this case, baby nipples were similarly applied over fistula openings and holes on top of the nipples served to place Malecot or Foley catheters to drain enteric fistula content. On the raw OA surface, petroleum jelly-impregnated gauze or clear Telfa sheet was placed and an abdominal wound VAC sponge was applied on top, subsequently receiving a negative pressure of 125 mmHg. Trevino et al. [161] reported the isolation of an EAF using a complex “ring” and “silo” to securely drain the fistula in a drainage collection bag, around which the VAC system was applied on the OA surface (ring and silo VAC). Fistulae were successfully isolated in 8 patients for a period of 3–20 weeks. Ramsay and Mejia [162] reported the successful diversion of the EAF after intubating them with Malecot catheters. ECFs were finally injected with fibrin glue and successfully sealed in all three patients.

The development of particular devices such as silicone fistula adapter, silicone distance mesh, polyvinyl alcohol sponge, and ABThera™ (KCI, Henderson, NV, USA) contributed to the technique safety and also avoiding wound contamination with enteral secretions (Figure 9).

## 10. Conclusions

In a nutshell, EFs usually affect patients in severe clinical conditions who cannot undergo a surgical operation because of their higher mortality and morbidity risks. These patients require a multimodal approach involving both total parenteral nutrition and NPWT. This association can reduce the fistula output and promote its healing, although with a longer treatment. It was proven that the closure rate of EAF is still not satisfactory, especially in high-output fistula. Thus, the general acceptance for the treatment of dominant EAFs with mucosal protrusion as a stoma is well-accepted worldwide. Patients with OA and EAFs have a high mortality rate. The surgical repair of an EAF has a high failure rate. Primary resection of the affected bowel loop results in the best outcome.

NPWT influences the spontaneous EF closure. This is true, especially for distal and low-output fistula. NPWT creates favorable conditions for the outflow of intestinal contents, enhances the granulation of the wound bed, and decreases local inflammation. Furthermore, NPWT avoids fistula recurrence after surgery.

Although significant progress in the field of NPWT for OA has been observed in recent years, there are still some technical aspects of the treatment that remain questionable with a lack of a firm consensus. Although there are some proposals for an algorithm for NPWT in OA with EAF, there is still an absence of solid recommendations. Thus, it is crucial to collect data and outcomes for creating guidelines regarding NPWT in OA management complicated with EAFs. The inadequacy of prospective studies and/or systematic reviews does not allow a proper comparison of different treatments. A patient-tailored approach is the best option currently.

## Figures and Tables

**Figure 1 jcm-13-01279-f001:**
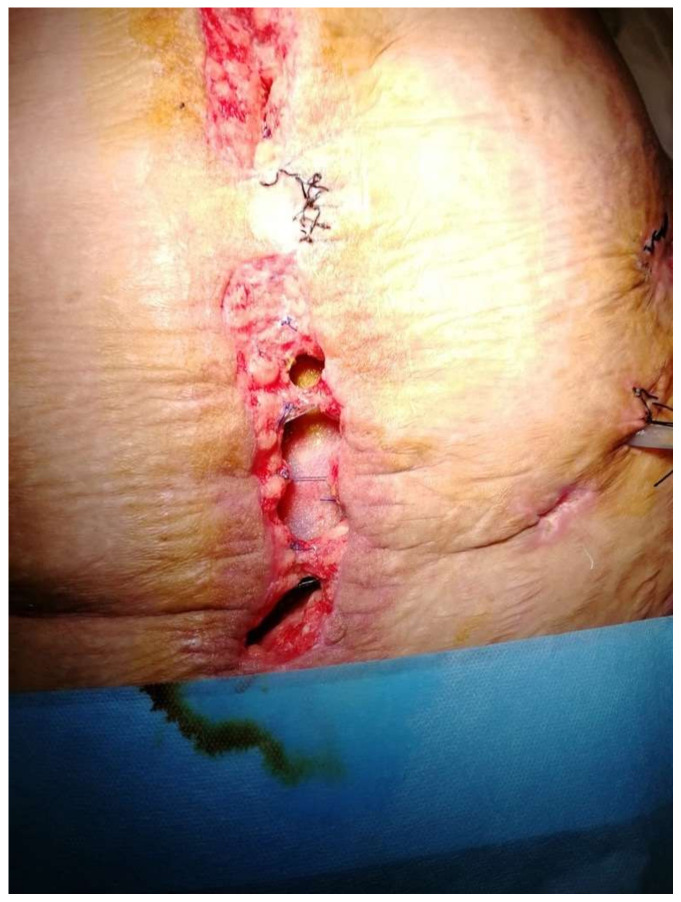
Enterocutaneous fistula. (Personal observation).

**Figure 2 jcm-13-01279-f002:**
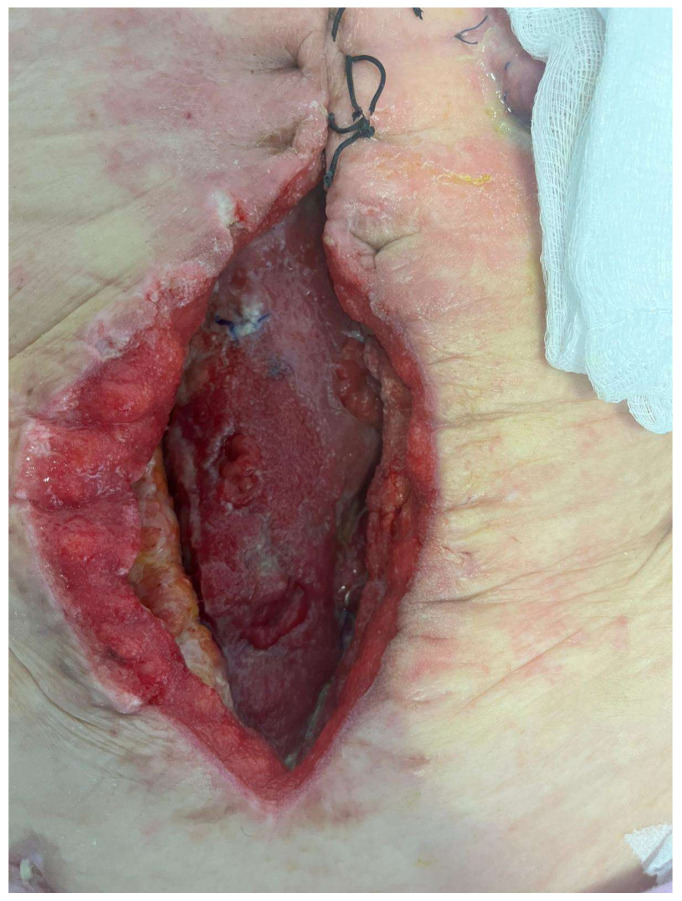
Entero-atmospheric fistula. (Personal observation).

**Figure 3 jcm-13-01279-f003:**
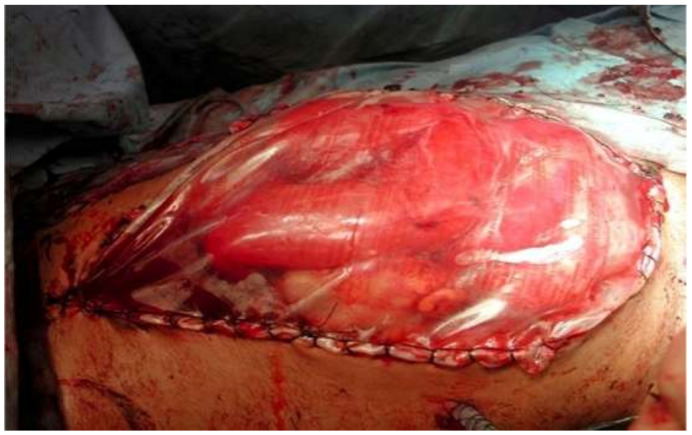
Damage control surgery for abdominal trauma. Open abdomen and Bogota bag (Personal observation).

**Figure 4 jcm-13-01279-f004:**
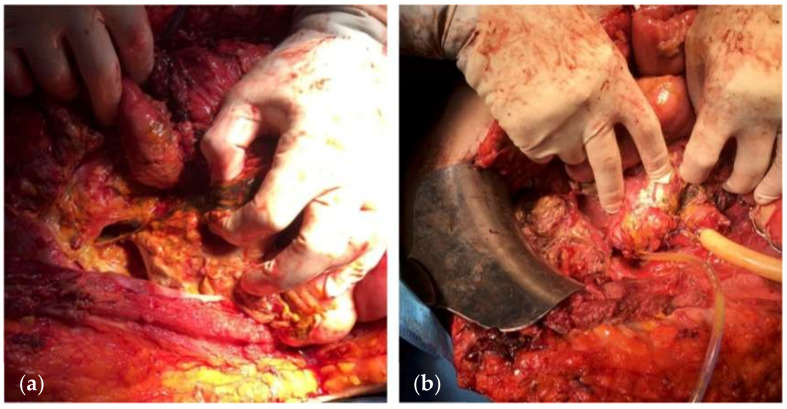
(**a**) Lateral duodenal fistula in a patient subjected to previous subtotal gastrectomy with Roux-en-Y reconstruction for gastrointestinal stromal tumor. (**b**) Reconstruction of the duodenal wall on a Petzer tube. Kehr T-drainage in the biliary tract. (Personal observation).

**Figure 5 jcm-13-01279-f005:**
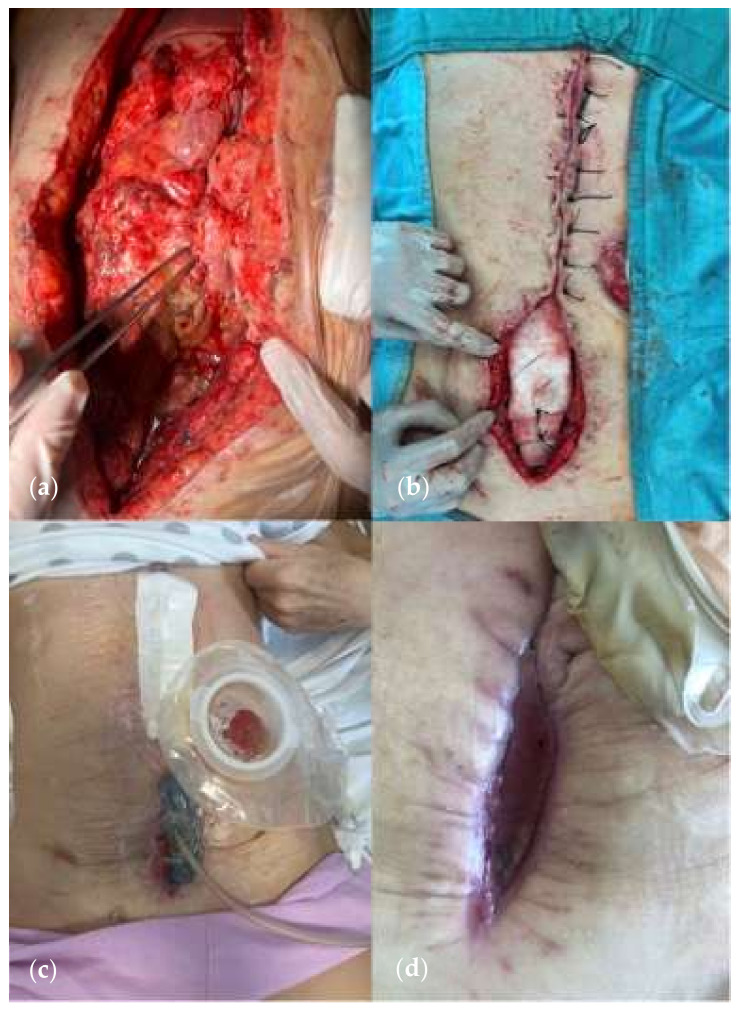
(**a**) A 65-year-old woman undergoing multiple surgical procedures with consequent various fistula formation. (**b**) A jejunostomy was performed with application of NPWT on white foam on the midline laparotomy. (**c**) NPWT on the midline incision 4 weeks after the first surgery. (**d**) Wound at 6 weeks. (Personal observation).

**Figure 6 jcm-13-01279-f006:**
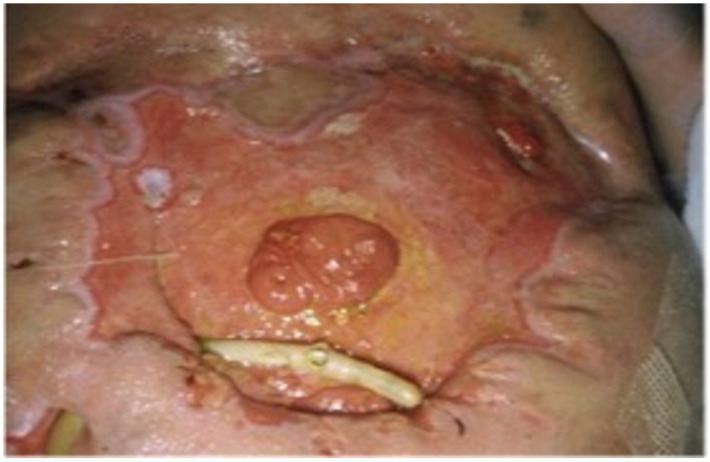
EAFs in hostile abdomen. (Personal observation).

**Figure 7 jcm-13-01279-f007:**
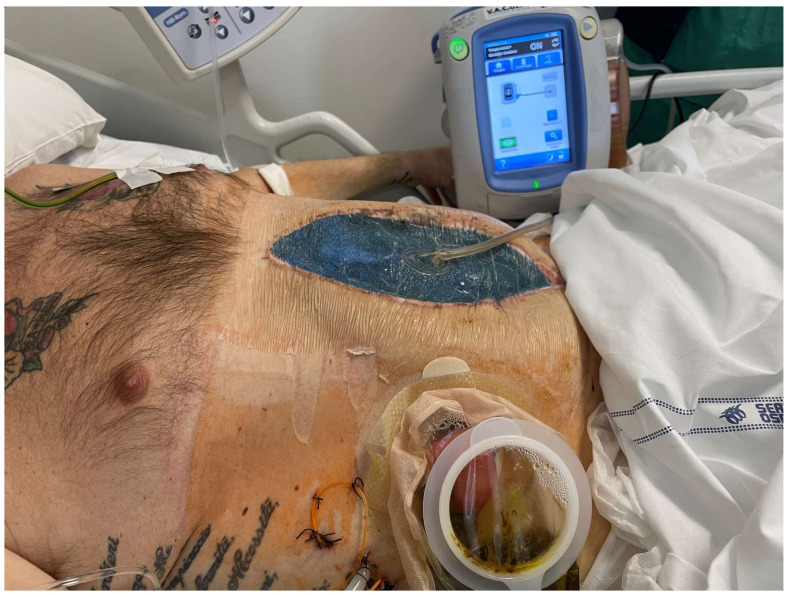
NPWT. (Personal observation).

**Figure 8 jcm-13-01279-f008:**
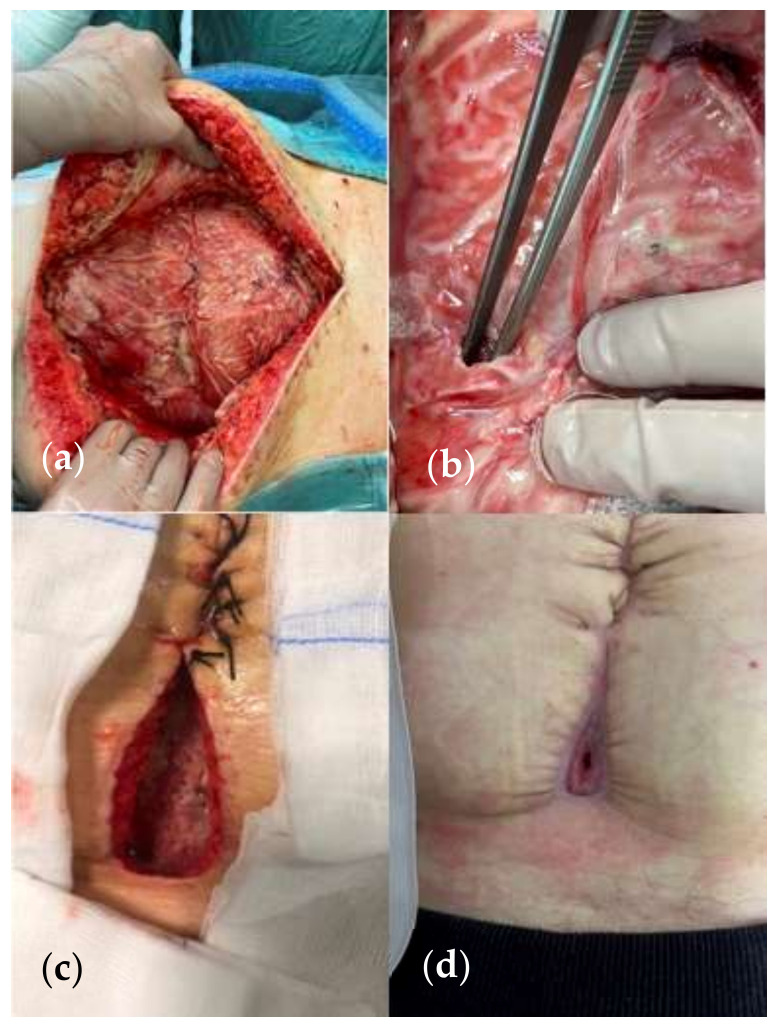
(**a**) A 36-year-old patient affected by multiple small bowel fistulas due to complications of ovarian cyst removal and consequent frozen abdomen. (**b**) EAF. (**c**) Treatment consisted of visceral NPWT with AbThera first and white foam later, allowing a progressive local improvement. (**d**) Injection of mesenchymal stem cells was performed at this point consenting a fistula and skin complete closure [142,143]. (Personal observation).

**Figure 9 jcm-13-01279-f009:**
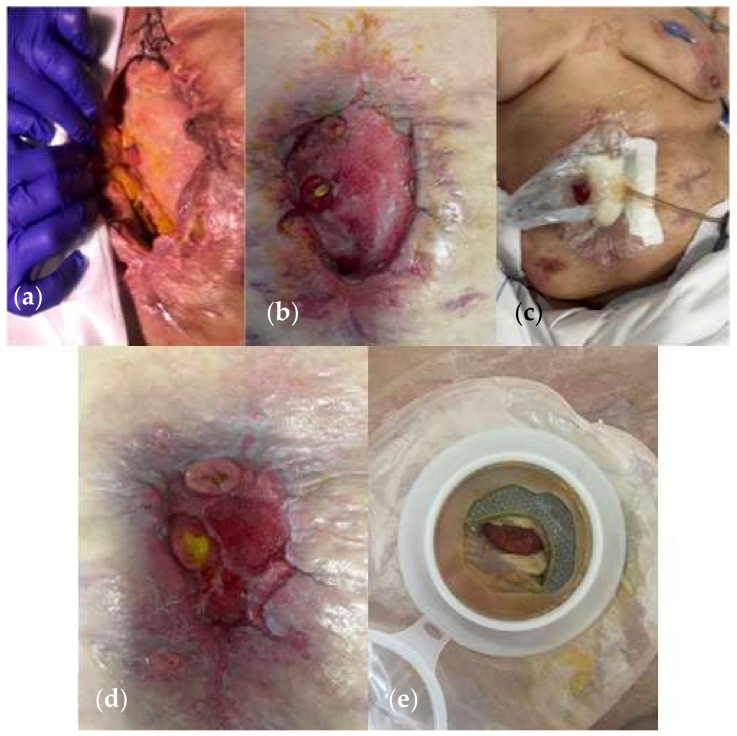
(**a**) A 54-year-old patient undergoing surgery for perforated diverticulitis with resection of the sigmoid colon and anastomosis. (**b**) Postoperative course complicated by dehiscence requiring multiple surgical re-explorations with consequent frozen abdomen and jejunal fistula. (**c**–**e**) She was treated with NPWT on white foam surrounding the fistula and a stoma bag connected to the machine through a large tube. Low negative pressure levels were set—50 mmHg. (Personal observation).

**Table 1 jcm-13-01279-t001:** Spontaneous closure based on site of origin, fistula characteristics, and environment and physiology.

Variables		Favorable	Unfavorable
Site of origin		Esophagus	Stomach
		Duodenal stump	Duodenum
		Pancreas	Proximal jejunum
		Biliary tree	Ileum
		Colon	
Characteristics and environment	Enteric defect	<1 cm	>1 cm
	Fistula tract	>3 cm	<3 cm
	Budding mucosa	Absent	Present
	Intestinal continuity	Intact	Disrupted
	Distal obstruction	Absent	Present
	Adjacent abscess	Absent	Present
	Bowel disease	Absent	Present
	Foreign body	Absent	Present
	Previous chemoradiation	No	Yes
Physiology	Fistula output	<500 mL/day	>500 mL/day
	Nutrition status	Well nourished	Malnourished
	Transferrin	>200 mg/dL	<200 mg/dL
	Sepsis	Absent/infrequent	Present/frequent

## Data Availability

All the data used are present in the text. No additional data are available.

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
