# Peer review of "Entero-Cutaneous and Entero-Atmospheric Fistulas: Insights into Management Using Negative Pressure Wound Therapy"

_jcm, 2024, doi:10.3390/jcm13051279_

Round 1
Reviewer 1 Report
Comments and Suggestions for Authors
Dear Authors,
I had the pleasure to review the paper "Enteric fistulas: insights in the management by negative pressure wound therapy."
I really enjoy your paper. The only suggestion I have is that one day you should insist in this paper and transform it in book chapter at least.
Congratulations
Author Response
My coauthors and I thank the reviewer for his appreciation of our manuscript. We thank the reviewer for the compliments he extended to us.
Reviewer 2 Report
Comments and Suggestions for Authors
It was a pleasure to review the manuscript, which is a thorough review of the intestinal fistula treatment. However, I have some reservations. The authors of the manuscript broadly describe the enteric fistulas, including the ones being complications after planned surgery, and describe all treatment options and not just the NPWT.
Considering the title (“Enteric fistulas: insights in the management by negative pres-2 sure wound therapy”), and the special issue topic (Minimally Invasive Emergency Surgery) I recommend a significant reduction of the text dealing with these topics and instead focusing on NPWT treatment and management of intestinal fistulas caused by trauma open abdomen treatment. In my opinion, the in-depth description of other treatment options and patients with intestinal fistulas caused e.g. by pancreatitis or by planned surgery complications is off-topic.
I would also recommend changing the first part of the title to “Entero-atmospheric fistulas: (…)” because these are the fistulas that the authors focus their text on.
Other concerns:
Line 172/173 please include a reference to the “recent experimental data” mentioned.
Line 167-68 and 177-178: These two statements seem contradictory: is the risk of EAF significantly increased after 2 weeks of OA treatment or is EAF most often seen after the first week of OA treatment? Please clarify in the manuscript, is the clinical observation of EAF formation after the first week the experience of the authors?
Line 226 Please explain exactly the “FRIEND” meaning (including which letter stands for which factor).
Line 276: Mentioning reconstructive surgery seems out of place here, it should be included at the end of the paragraph on fistula treatment.
Line 329: the choice of the right mesh is mentioned twice. Please correct.
In the paragraph starting with line 328 the authors mention operating techniques appropriate for patients with a well-managed fistula. I recommend focusing on the NPWT treatment and only shortly mentioning other treatment options, which usually may be applied after successful NPWT.
Line 407: “It is recommended to set a lower suction pressure” – please include reference and the suggested pressures; “lower” is not accurate enough.
Lines 416-448: The authors include multiple NPWT studies not directly concerned with the manuscript topic. Please abbreviate the paragraph and include general information on the NPWT technique, focusing on its use for intestinal fistulas.
Please review the references, there are at least 6 self-citations (No 2,8,9,38,39,51), not necessarily concerned with the main topic (I suggest choosing other sources on fistula etiology).
Comments on the Quality of English Language
The manuscript is well written, there are very few minor errors to correct.
Author Response
My coauthors and I thank the reviewer for his appreciation of our manuscript.
The manuscript has been modified in light of the reviewer's comments. Text changes have been highlighted in yellow.
The title has been changed. In the manuscript we write not only about entero-atmospheric fistulas but also about entero-cutaneous fistulas. Our manuscript is not focused on entero-atmospheric fistulas alone after open abdominal treatment.
The sentence on line 172-173 has been changed.
The sentences on lines 167-168 and 177-178 have been changed. Edited text is highlighted in yellow.
The acronym FRIEND on line 226 has been removed.
The text on line 276 on reconstructive surgery has been changed and moved to the end of the paragraph on fistula treatment.
The text on line 329 has been corrected. The fix is highlighted in yellow.
The paragraph starting at line 328 has been changed.
The text on line 407 has been changed.
The list of references has been modified.
Reviewer 3 Report
Comments and Suggestions for Authors
Enteric fistulas are a complex issue in abdominal and gastrointestinal surgery and are linked to considerable rates of illness and death. This is owing to complications such as sepsis, starvation, and electrolyte imbalance. The theme is very well chose because is it very important for surgeons.
I have a series of comments:
The text is very well written and structured. It still needs small language adjustments because there are too long phrases that are difficult to follow and understand, but otherwise everything is fine.
In terms of importance, the study touches on an important problem encountered in abdominal surgical practice, and what it sheds light on could help treat these unwanted complications.
The study is well structured and results from other studies are compared in the discussions.
I would be interested to see Prisma to highlight the way of choosing the studies in table 2 and also if a comparison can be made with the authors' own case studies in this field.
In addition, the consistent bibliography is worth noting, which underlines the authors' interest in this topic.
Comments on the Quality of English Language
The text is very well written and structured. It still needs small language adjustments because there are too long phrases that are difficult to follow and understand, but otherwise everything is fine.
Author Response
My coauthors and I thank the reviewer for his appreciation of our manuscript.
We thank the reviewer for the compliments he extended to us.The manuscript has been revised. Text changes have been highlighted. The manuscript was written in English by Dr. Maria Michela Chiarello, who worked for two years in Edinburgh, at a prestigious hospital. For this reason, my co-authors and I did not submit the manuscript to an external proofreading service.
Round 2
Reviewer 2 Report
Comments and Suggestions for Authors
Dear Authors,
Thank you for the thorough revision. I find the article is now well structured and appropriate for publication.